# Current Nutritional and Pharmacological Approaches for Attenuating Sarcopenia

**DOI:** 10.3390/cells12192422

**Published:** 2023-10-09

**Authors:** Kunihiro Sakuma, Kento Hamada, Akihiko Yamaguchi, Wataru Aoi

**Affiliations:** 1Institute for Liberal Arts, Environment and Society, Tokyo Institute of Technology, Meguro-ku, Tokyo 152-8550, Japan; hamada.k.ae@m.titech.ac.jp; 2Department of Physical Therapy, Health Sciences University of Hokkaido, Kanazawa, Ishikari-Tobetsu, Hokkaido 061-0293, Japan; yama@hoku-iryo-u.ac.jp; 3Laboratory of Nutrition Science, Graduate School of Life and Environmental Sciences, Kyoto Prefectural University, Kyoto 606-8522, Japan; waoi@kpu.ac.jp

**Keywords:** sarcopenia, HMB, catechin, ursolic acid, vitamin D, myostatin, muscle atrophy

## Abstract

Sarcopenia is characterized by a gradual slowing of movement due to loss of muscle mass and quality, decreased power and strength, increased risk of injury from falls, and often weakness. This review will focus on recent research trends in nutritional and pharmacological approaches to controlling sarcopenia. Because nutritional studies in humans are fairly limited, this paper includes many results from nutritional studies in mammals. The combination of resistance training with supplements containing amino acids is the gold standard for preventing sarcopenia. Amino acid (HMB) supplementation alone has no significant effect on muscle strength or muscle mass in sarcopenia, but the combination of HMB and exercise (whole body vibration stimulation) is likely to be effective. Tea catechins, soy isoflavones, and ursolic acid are interesting candidates for reducing sarcopenia, but both more detailed basic research on this treatment and clinical studies in humans are needed. Vitamin D supplementation has been shown not to improve sarcopenia in elderly individuals who are not vitamin D-deficient. Myostatin inhibitory drugs have been tried in many neuromuscular diseases, but increases in muscle mass and strength are less likely to be expected. Validation of myostatin inhibitory antibodies in patients with sarcopenia has been positive, but excessive expectations are not warranted.

## 1. Introduction

Skeletal muscle contraction is the driving force behind physical movement and plays an important role in homeostasis. Sarcopenia occurs as a consequence of age-related loss of muscle strength, quality and quantity. It often refers to cellular processes (mitochondrial dysfunction, inflammation, and denervation) and the resulting loss of muscle strength, function and mobility, and increased risk of falls. Sarcopenia is considered ‘primary’ (or age-related) when there is no apparent cause other than age [1]. Secondary sarcopenia usually occurs when multiple causes are present. This condition is commonly observed in the elderly and is due to diabetes, stroke, hip fracture, chronic obstructive pulmonary disease, and chronic heart failure [2]. Muscle mass loss is more pronounced in lower limb muscle groups than in upper limb muscle groups and, at the myofiber level, is characterized by selective type II muscle fiber atrophy and fibronecrosis.

The mechanism of age-related muscle atrophy is thought to be influenced simultaneously by several factors, including poor muscle regeneration, hormonal changes, nutritional deficiencies, and increased oxidative stress. Although the specific contribution of these factors remains to be elucidated, it has been reported that muscle hypertrophy regulators become less responsive with age [3], and dysfunction of autophagy is also evident [4,5]. In contrast, no apparent changes in negative regulators (atrophy gene-1 (atrogin-1), myostatin, NF-κB (nuclear factor-κB)) have occurred in ageing mammalian muscle [6,7]. Strategies such as nutritional supplementation and physical training (both aerobic and resistance exercise) are important interventions for muscle atrophy in the elderly.

Dietary protein provides amino acids necessary for muscle protein synthesis and, as an anabolic stimulus, acts directly on protein synthesis. There is a considerable literature on the role of mixtures containing all essential amino acids (EAAs) in muscle metabolism and prevention of sarcopenia, both in experimental models and in humans [8,9]. The administration of multiple essential amino acids also increases muscle mass and protein synthesis, even under normal conditions without resistance training [8]. In addition, a number of reviews suggest that nutritional interventions such as high protein intake and intake of leucine, a branched-chain amino acid, combined with resistance training would be effective in reducing muscle fiber atrophy in sarcopenia [10,11]. On the other hand, mild calorie restriction (CR) attenuates sarcopenia in all mammals quite effectively [12,13]. Recent studies have also indicated that new supplements (e.g., ursolic acid, resveratrol, and soy isoflavones) could be applied to prevent muscle atrophy [14,15]. Furthermore, despite many negative reports, pharmacological sarcopenia treatment strategies have also attracted the attention of many researchers. This chapter details recent nutritional and pharmacological approaches to control sarcopenia using the database of PubMed.

## 2. Nutritional Approach

### 2.1. HMB

Protein supplementation/a protein-rich diet is conditionally recommended by clinical practice guidelines for sarcopenia based on the International Conference on Sarcopenia and Frailty Research (ICFSR) [16]. In addition, in the consensus statements of Asian Working Group for Sarcopenia, high-quality protein, amino acids such as leucine and L-carnitine, or oral nutritional supplement (ONS) containing beta-hydroxy-beta-methylbutyrate (HMB) may be considered and should be taken according to the specific prescribing information [17]. HMB is a metabolite of leucine, a branched-chain amino acid, and HMB stimulates protein synthesis via mTOR and decreases myonuclear apoptosis and proteasome expression. Supplementation with HMB is demonstrated to increase fat oxidation and mitochondrial biogenesis [18]. The effects of HMB supplementation in humans has been examined by many researchers. Deutz et al. [19] reported positive results using 10 days of HMB (3 g) supplementation for unloading patients (bed rest). Specifically, HMB intake prevented muscle weakness during unloading and maintained muscle strength during the rehabilitation period. The HMB intake used in this study was a dose commonly used in academic research, and the subjects were diet-controlled. In addition, Hsieh et al. [20] observed the positive effects of HMB intake (a decrease in blood urea and urinary urea excretion) during a 2-to-4-week intervention period. Notably, they suggested a decrease in proteolysis despite no significant change in anthropometric parameters due to HMB supplementation. However, their study had some problems, including a short duration of supplementation. In contrast, a meta-analysis of several randomized controlled trials (RCTs) showed that HMB supplementation for the elderly prevented loss of lean body mass without changing fat mass, and HMB may be more effective by combining it with other nutrients such as lysine and arginine. Specifically, 12 weeks of HMB/Arg/Lys mixture (2/5/1.5 g per day) has been reported to increase whole body protein synthesis, leg and grip strength, and limb circumference [21]. However, their study found no increase in muscle strength, thus negating the effect of HMB supplementation on muscle strength. Furthermore, a high-quality systematic review suggested no significant effect of HMB supplementation on muscle strength [22]. However, this meta-analysis showed a great deal of evidence suggesting the effect of HMB supplementation on muscle mass. In summary, HMB supplementation for older adults has a positive effect on muscle mass but has no clear effect on physical performance and muscle strength. From the two reviews, where evidence is somewhat lacking, it has not been determined whether HMB alone is effective in improving muscle strength, physical performance, or muscle mass [22], and further research is needed regarding the effects of HMB supplementation on reducing sarcopenia.

Whether HMB alone improves symptoms of sarcopenia has not been clearly elucidated, but HMB and vibration stimulation appear to significantly improve sarcopenia. Whole-body and local vibration are often used to improve muscle weakness in the elderly. An RCT of men with sarcopenia (mean age 88.6 years) showed that vibration (12–16 Hz, 3–5 mm) for 8 weeks significantly improved muscle strength and physical performance (walking speed, 5-stance test, and timed stand test) [23]. Although systematic reviews and meta-analyses have shown that vibration stimulation therapy is an effective strategy to improve physical performance and muscle strength in patients with sarcopenia, caution is necessary when interpreting the results because of the limited number of publications on this topic. Wang et al. [24] recently showed that LMHFV significantly improves skeletal muscle quality by suppressing myostatin expression and reducing intramuscular fat mass in aging mice (SAMP8). In particular, vibration stimulation combined with HMB significantly increased lean body mass and improved muscle performance (single contraction and muscle strength).

### 2.2. Polyphenol

#### 2.2.1. Catechin

Tea (*Camellia sinesis*), especially green tea, which is primarily composed of catechins, contains many phenolic hydroxyl groups (-OH). There are four monomers of catechins: epigallocatechin gallate (EGCG), epicatechin (EC), epigallocatechin (EGC), and epicatechin gallate (ECG). Green tea catechins have been shown to act on skeletal muscle cells and may inhibit muscle mass loss. Supplementation with EGCG (5 mg/kg, 4 times/week) for 8 weeks appears to reduce fibrosis and necrotic muscle fibers in muscular dystrophic mice [25]. EGCG supplementation also decreases protein carbonyl content (an oxidative stress marker) in aged male rats [26]. However, some reports suggest that the effects on muscle adaptation depend on the type of catechin being supplemented. In an experiment in which 20-month-old male mice were fed EC or EGCG (0.25% in drinking water) and a standard diet, survival significantly increased from 39% to 69% after 37 weeks of EC consumption, whereas no significant effect was observed with EGCG [27]. In addition, EC slowed skeletal muscle degeneration and improved physical activity in aged mice, indicating that the effects of EC on skeletal muscle may be more pronounced than those of EGCG.

A recent double-blind, controlled trial was performed to examine the effects of green tea in humans using 62 men. They randomized four groups to receive epicatechin for 8 weeks: resistance training (RT), 1 mg of epicatechin (EC) per kg of body weight, resistance training plus epicatechin (RT + EC), and placebo (PL). The results showed significant increases in follistatin, follistatin/myostatin ratio, and muscle strength (leg press and chest press) in the RT + EC group compared to those in the other three groups. In a randomized controlled trial, 128 women with sarcopenia were randomly assigned to four groups: exercise and tea catechin intake (350 mL/day), exercise, tea catechin supplementation, and 3 months of health education. After the intervention, a significant effect on the composite variables of leg muscle mass and normal walking speed was observed in the exercise and catechin intake group compared the health education group [28].

Lychee extract (oligonol), rich in phenolic compounds (flavonols), prevents kidney damage caused by diabetes and obesity caused by a high-fat diet [29]. The addition of oligonol decreased the expression of PGC-1α and Mfn2 and increased the expression of LC3-II, p62, and ATG13, and showed autophagosome and lysosome accumulation [30]. Although numerous studies have investigated the effects of green tea extract supplementation (EC, EGCG, oligonol, etc.) on mitochondria and muscle fibers, almost all studies have been conducted in experimental animals. The safety, bioavailability, and efficacy of green tea should be tested in patients with sarcopenia. There is insufficient evidence for the effect of catechins on sarcopenia patients, as most clinical studies examining the effect of green tea extract in sarcopenia patients have not been RCT studies.

#### 2.2.2. Isoflavones

Isoflavones, a type of flavonoid found in soybeans, are structurally similar to estrogens and exert their physiological functions, such as antioxidant effects, by binding to estrogen receptors. Estrogen receptors (ERs) include ERα and ERβ, and daidzein, a soy isoflavone, acts on ERβ, but not ERα. In post-oophorectomy mice, muscle mass increases when isoflavones are added to a high-fat diet for 12 weeks [31]. In contrast, long-term (120 days) isoflavone intake decreases fat accumulation in the skeletal muscle of male mice [32]. Notably, isoflavones affect muscle mass and function in both male and female mice. Many researchers examined the effects of isoflavones on muscle mass but have not observed the changes in muscle fiber size. Moreover, accumulating large amounts of fat and connective tissue in atrophied tissue may render muscle weight-based assessment methods inaccurate. Abe et al. [33] showed that isoflavone administration (20% of the diet) significantly reduced muscle fiber atrophy in the tibialis anterior muscle on day 4 after denervation. Furthermore, isoflavone treatment caused increased expression of IRS-1 and p-Akt1 proteins in mouse denervated muscle.

Most in vivo isoflavone supplementation experiments have used isoflavones in amounts greater than 1% of dietary intake [31,32], but it is difficult for humans to consume such large amounts of isoflavones in every meal. Therefore, testing the effects of isoflavones beyond 1% of dietary intake may be meaningless. Our group found that administration of 0.6% isoflavone (AglyMax, an isoflavone aglycon mixture of daidzein, genistin, and glycitein in a 7:1:2 ratio) inhibited muscle fiber atrophy after denervation in mice [34]. This effect is thought to be due to a reduction in apoptosis-dependent signaling rather than atrogin-1 pathway-dependent signaling.

Intake of daidzein may be beneficial for older women. For example, 8-prenylnaringenin has estrogenic effects similar to daidzein and appears to inhibit denervation-induced skeletal muscle atrophy by promoting Akt phosphorylation. Quercetin, a type of flavonoid, is also abundant in vegetables and fruits and has a scavenging effect on reactive oxygen species (ROS). Since ROS activate NFκB and Foxo pathways and induce the expression of E3 ubiquitin ligase, quercetin’s ability to suppress muscle atrophy has attracted attention. For example, the ingestion of quercetin in hindlimb-unloaded mice reduces the expression of atrogin-1 and MuRF-1 in gastrocnemius muscle and prevents the loss of skeletal muscle mass [35]. Future studies are required to elucidate whether daidzein, quercetin, and AglyMax inhibit muscle aging.

### 2.3. Ursolic Acid (UA)

Ursolic acid (UA) is a natural pentacyclic triterpenoid carboxylic acid found in plants and fruits such as apples, blueberry and cranberry. UA has anti-carcinogenic, antioxidant, anti-inflammatory, and anti-obesity effects, and UA has also been shown to have beneficial effects in animal models of hyperlipidemia and diabetes. A systematic review involving animal studies concluded that the supplementation with UA increases muscle strength and muscle mass by increasing mTOR pathway activation in muscle satellite cells. UA has been reported to reduce the expression levels of two muscle atrophy genes (atrogin-1 and MuRF1) and reduce muscle atrophy caused by fasting and atrophy-induced stress [36]. UA supplementation also increases Akt phosphorylation in skeletal muscle in vivo but UA may not act directly on skeletal muscle because UA alone is not sufficient to promote IGF-I and insulin receptor activation. Yu et al. [37] orally administered UA (100 mg/kg) to mice models of chronic kidney disease (CKD) for 3 weeks. They found that UA administration reduced the expression of inflammatory cytokines (e.g., IL-6 and TNF-α) and ubiquitin E3 ligases (e.g., MuRF-1, atrogin-1, and MUSA1) and suppressed CKD-induced muscle atrophy (tibialis anterior muscle).

Moreover, supplementation with UA is highly effective for improving muscle tissue in combination with resistance and endurance exercise. Endurance training and UA (ET + UA) decreased the body weight, insulin resistance, and insulin and glucose concentrations more than the other groups [38]. Furthermore, the ATF4 protein level was significantly reduced in ET + UA compared with DM. ET + UA reduced the amount of p53 protein similar to the significant decrease in p21 protein, which was diminished more than that in any other group. In contrast, the amount of p21 in the RT + UA group was not more significant than that in UA group. Collectively, these findings show that exercise and UA supplementation impeded the interactions among p53/ATF4/p21.

The effects of UA intake on skeletal muscle have not been well tested in humans. UA supplementation (loquat leaf extract intake, 50 mg/day) in healthy adults did not significantly improve muscle mass and strength [39]. In addition, the combination of strength training and 8 weeks of UA supplementation increased muscle strength but not lean body mass compared with strength training alone [40]. Further studies are needed to clarify the effects of UA supplementation on sarcopenia in humans.

### 2.4. Omega-3 PUFAs

Lipids such as omega-3 fatty acids and n-3 polyunsaturated fatty acids have been suggested to affect muscle health indirectly by preventing low-grade inflammation [41]. Intriguingly, significant lower levels of serum 3 fatty acids are associated with sarcopenia, higher dietary lipid intake correlates with lower odds of sarcopenia [42]. More recently, AWGS consensus guidelines found scarce evidence to support a statement around dietary/supplementary lipid intake in muscle health [17]. Omega-3 PUFAs include alpha-linolenic acid (ALA), eicosapentaenoic acid (EPA), docosahexaenoic acid (DHA), and docosapentaenoic acid (DPA). Concentrations of DHA and EPA in cell membranes are important to ensure structural cell function. The primary dietary sources of DHA and EPA are seafood and fatty fish (e.g., mackerel). ALA is found in plant foods such as vegetable oils, chia seeds, and nuts. Although 500 mg of DHA and EPA are recommended daily, most U.S. adults do not achieve this intake. Omega-3 PUFAs are of great interest because of their anti-inflammatory properties and low risk of adverse events. Thus, supplementation with omega-3 PUFAs is beneficial for a variety of aging processes, including cognition, eye health, and bone metabolism. Omega-3 PUFAs may also have beneficial effects in muscles, including hypertrophy of muscle fiber and improvement of mitochondrial function. Furthermore, RCT studies have indicated that supplementation with omega-3 PUFAs promotes muscle protein synthesis in the elderly [43]. The relationship between dietary intake of omega-3 PUFAs and physical performance has also been studied in a cross-sectional study: 2893 men and women (aged 59–73 years) included in the Hertfordshire Cohort study were examined to determine the association between grip strength and dietary intake. The study showed a relationship between each additional serving of fatty fish and increased grip strength in both sexes [44].

The anti-inflammatory effects of omega-3 PUFAs are generally accepted. Supplementation of omega-3 PUFAs in middle-aged and older adults causes significant IL6 and CRP reductions. In addition, an RCT study with elderly subjects showed that treatment with EPA and DHA (for 8 weeks) significantly reduced blood levels of TNFα, IL-6, and IL-1β [45]. Suppression of inflammation by omega-3 PUFAs may be one mechanism to combat sarcopenia because chronic low-level inflammation is involved in its development. In contrast, there are reports stating that supplementation with omega-3 PUFAs does not affect muscle protein synthesis [46], and further studies are needed in elderly subjects with sarcopenia. Moreover, Smith et al. found more amino acid and insulin-stimulated activation of the mTOR-p70S6K pathway after 8 weeks of omega-3 PUFA supplementation compared with a placebo group. Omega-3 PUFA supplementation also produces important changes in gene expression that promote skeletal muscle anabolism by decreasing the expression of the mTOR inhibitory pathway. Thus, omega-3 PUFAs have the potential to overcome age-related anabolic resistance through the activation of mTOR signaling. Figure 1 represents an overview of nutritional intervention for sarcopenia. Table 1 summarizes the nutritional approaches for attenuating muscle atrophy.

## 3. Pharmacological Approach

ICFSR clinical practice guidelines indicate that pharmacological interventions are not recommended as the first-line therapy for the management of sarcopenia [16].

### 3.1. Vitamin D

Vitamin D is an important regulator of bone metabolism, including phosphorus and calcium homeostasis. Vitamin D deficiency and loss of muscle mass often occur together; it is estimated that approximately 50% of individuals over 65 years of age have hypovitaminosis D levels. Although the mechanism remains unclear, vitamin D appears to influence muscle function and strength through the vitamin D receptor. Notably, the expression of vitamin D receptors in skeletal muscle decreases with age.

A double-blind study of nursing home residents and older adults living in areas with low vitamin D has shown that supplementation with vitamin D reduces the risk of falls and improves muscle performance and strength [47]. Many epidemiological studies have also shown that lower extremity function is impaired in elderly patients with hypovitaminosis D status, including delayed walking time and time from sitting to standing [48]. To prevent sarcopenia, it may be useful to clarify the effects of vitamin D on skeletal muscle function, strength, and muscle mass. However, a recent review [49] using eligible RCTs clearly denied the positive effect of vitamin D monotherapy in older adults (age 50 years and older). A random-effects inverse-variable model was used to calculate the mean difference between groups for handgrip strength (HGS), timed up-and-go (TUG), and appendicular lean mass (ALM). Significant reductions in short physical performance battery scores were observed for vitamin D supplementation compared with the placebo. Vitamin D supplementation did not affect TUG, HGS, ALM, general muscle strength, or general physical performance. Therefore, vitamin D supplementation has been shown not to improve sarcopenia in elderly individuals who are not vitamin D-deficient. This conclusion would be consistent with the ICFSR clinical practice guideline that there is insufficient evidence to determine whether vitamin D replacement therapy is effective in elderly patients with sarcopenia [16].

### 3.2. Myostatin Inhibition

Myostatin, a novel member of the transforming growth factor beta superfamily, negatively regulates muscle growth [50]. Myostatin expression in muscle is clearly increased in CKD mice and rats [51,52]. However, conflicting results have been obtained as to whether myostatin expression is increased in sarcopenic muscle [6,53]. Many investigators have evaluated the effects of myostatin inhibition in muscle disease models such as cancer cachexia, CKD, Duchenne muscular dystrophy (DMD), and amyotrophic lateral sclerosis [54]. For example. subcutaneous injection of anti-myostatin peptides in CKD mice restored weight loss (5–7% increase) and muscle mass loss (~10% increase) in the hindlimbs, including the TA, gastrocnemius, EDL, and soleus muscles. Myostatin inhibition also suppressed circulating inflammatory cytokines (TNF-α, M-CSF-1, etc.) and the mRNA expression of these cytokines in CKD mice.

The effects of pharmacological inhibition of myostatin in DMD patients have been examined by several researchers but positive effects have been difficult to obtain and have little potential for clinical application. Furthermore, a randomized clinical trial of the same compound in DMD patients showed improved muscle mass and performance (6 min walk test), but was stopped early owing to non-muscular side effects (nosebleeds, capillary dilation, etc.) [55]. LY2495655, a phase II humanized antibody against myostatin, showed a dose-dependent and significant increase in lean mass compared with the placebo at weeks 8 and 16 after an RCT of total hip replacements in 400 patients [56]. Becker et al. [57] conducted a randomized phase II trial for the subcutaneous injection of a myostatin antibody (LY2495655: LY, 315 mg) in frail elderly subjects worldwide (e.g., the US, France, and Australia), aged 75 years and older. After 24 weeks, the LY group showed a significant increase in the least-squares mean change in aLBM compared with the placebo group. In addition, functional characteristics, such as stair climbing time and chair standing with arms, improved from baseline after 24 weeks of LY treatment in elderly subjects [57]. Based on these results, antibody inhibition of myostatin may be effective in treating sarcopenia, but the results of many clinical trials do not support this notion. Although beneficial effects in terms of fat loss and glucose metabolism have been consistently observed in many clinical trials with antimyostatin drugs, many have been disappointing in terms of increased muscle mass and functional improvement [58].

Researchers are particularly interested in the role of GDF11, which bears many similarities to myostatin in terms of the amino acid sequence and receptor and signaling pathways. For example, GDF11 has been reported to inhibit myocardial hypertrophy and decrease with age [59]. Additionally, supplementation of aged mice with GDF11 had a positive effect on aging satellite cells and improved skeletal muscle regeneration [60]. Some of the early studies reported the rejuvenating effects of GDF11 in aging mice and suggested that GDF11 can restore the skeletal muscle, heart tissue, and cerebrovascular system to a young and healthy state [61]. However, many subsequent studies have shown that blood GDF11 levels are either unaffected by aging or increase, with contradicts the previous studies.

Early studies also suggested that GDF11, like myostatin, inhibits satellite cell proliferation, induces differentiation, and inhibits muscle regeneration. However, increased GDF11 was found to be a risk factor for frailty and cardiovascular disease [61], demonstrating that exogenous GDF11 induces skeletal and cardiac muscle decline in mice. These contradictory findings regarding the role of GDF11 may be the result of using *E. coli* to create the recombinant GDF11 protein [59,60]. *E. coli* does not have an oxidative environment and thus cannot form the disulfide bond that connects the two GDF11 monomers. The use of GDF11 recombinant protein produced in *E. coli* in experiments with proper physiological function verification may lead to more reliable results.

### 3.3. Anabolic Steroids

The current evidence is insufficient to recommend anabolic hormones for the management of sarcopenia in ICFSR consensus guidelines [16]. Testosterone levels in men decrease by 1% per year after age 30, and bioavailable testosterone levels decrease by 2% per year [62]. Blood levels of testosterone in women decline rapidly between the ages of 20 and 45. With renal impairment (progression of CKD stage), testosterone levels markedly decrease [63], and the decrease in testosterone levels in CKD patients is significantly correlated with levels of sperm cytoplasmic droplets and total neutral glucosidase activity [63]. A systematic review [64] has shown that testosterone supplementation attenuates sarcopenia characteristics such as decreased muscle mass [65] and grip strength [66]. In a placebo-controlled study of long-term (6 months) supraphysiologic testosterone administration, leg lean body mass and leg and arm muscle strength increased [67]. Older men receiving high doses of testosterone show significant increases in muscle strength, but the potential risks outweigh the benefits. Increased risk of thrombotic complications, sleep apnea, and prostate cancer are associated with testosterone therapy in older men.

New nonsteroidal compounds called selective androgen receptor modulators SARMs exhibit tissue-selective activity, improved pharmacokinetic properties, and fewer systemic side effects. The effects of SARMs have been tested in healthy older men and patients in cancer-related sarcopenia [68,69]. In fact, a phase II double-blind study of enobosam, a nonsteroidal SARM, showed improvements in physical function (stair climbing speed) and LBM in healthy postmenopausal elderly men [70]. This study showed that as little as 1 mg of enobosarm increased LBM in patients with advanced cancer. Another SARM, GSK2881078, administered once daily for 50 days to healthy elderly men and women, significantly increased lean body mass without significant side effects, although alanine aminotransferase was transiently elevated [70]. More rigorous clinical trials are necessary to determine whether these agents are effective in normal and CKD-induced sarcopenia.

Supasyndh et al. [71] evaluated supplementation with oxymetholone, a milder anabolic steroid, in a 24-week RCT with 43 hemodialysis patients and observed significant increases in fat-free volume, subjective scores of physical function, and grip strength. Hemodialysis patients showed atrophy of type I and type II muscle fibers and decreased IGF-IR mRNA levels after 24 weeks of placebo treatment, but this was suppressed by oxymetholone, which induced hypertrophy of type I muscle fibers. Recently, Kim et al. studied the effects of oxymetholone (50 mg/kg) on old mice after 28 days of chronic forced exercise [72]. Glutathione, superoxide dismutase, and ROS were partially abundant owing to decreased catalase activity, but were restored to baseline levels by oxymetholone supplementation. In addition, oxymetholone treatment suppressed the expression of myostatin, sirtuin 1, and muscle atrophy genes and induced fiber hypertrophy in the gastrocnemius and soleus muscles [72]. Thus, there are few reports of oxymetholone treating sarcopenia or CKD in mouse models or in humans. These results should be clarified to assess the benefits and risks to sarcopenia due to normal or CKD with treatment with anabolic steroids, androgens, or similar substances.

### 3.4. Glucose-Lowering Drugs

#### 3.4.1. Metformin

Metformin is the most commonly prescribed antidiabetic drug and has been reported to activate AMPK and stimulate PGC1α gene expression in skeletal muscle. This promotes the transcription of genes associated with fatty acid oxidation and suppresses intramuscular fat accumulation. Activation of the AMPK pathway also promotes angiogenesis, mitochondrial biosynthesis, and the conversion of myofiber types from glycolytic to oxidative fibers [73,74,75]. Furthermore, it has been reported that Metformin suppresses the expression of mTORC1-related genes in the skeletal muscle of elderly individuals with impaired glucose tolerance [76]. Since autophagy is dysfunctional in sarcopenia, activating AMPK upstream of the autophagy pathway and suppressing mTORC1 may promote metabolism, resulting in an anti-aging effect.

Clinical trials have shown that Metformin treatment (3 × 500 mg for 16 weeks) in elderly patients with diabetes improved walking speed by 0.13 m/s [77]. Because most of the reports from clinical trials have been in patients with diabetes, there is insufficient evidence for Metformin’s treatment of sarcopenia. However, it may have a therapeutic effect through the aforementioned signaling pathways.

#### 3.4.2. Thiazolidinediones (TZDs)

TZDs increase the expression of adiponectin and its receptors in the blood and increase the expression of genes related to mitochondrial function and fatty acid oxidation by activating AMPK and ACC in skeletal muscle [78]. This results in increased skeletal muscle insulin sensitivity [79]. In a clinical trial in non-diabetic patients, men treated with pioglitazone showed a decrease in thigh muscle mass compared with men who did not receive pioglitazone [80]. TZDs may not be effective enough for the treatment of sarcopenia.

#### 3.4.3. Sulfonylureas

Sulfonylurea acts on pancreatic beta cells to promote insulin secretion and has been reported to decrease muscle protein in rat flexor digitorum brevis muscle through caspase-3-dependent or -independent pathways in in vitro experiments [81]. In a cross-sectional study in diabetic patients, sarcopenia parameters (skeletal muscle mass, skeletal muscle mass index (SMI), muscle strength, and gait speed) were reported to worsen more with sulfonylureas compared with DPP-4i [82], and there were no changes in skeletal muscle mass in diabetic patients in clinical trials [83]. The effect on sarcopenia is not clear, but likely does not contribute to its improvement.

#### 3.4.4. Dipeptidyl Peptidase-4 Inhibitor (DPP-4i)

DPP-4i promotes insulin secretion and inhibits glucagon release. In skeletal muscle, DPP-4i has been reported to increase GLUT4 expression in soleus and gastrocnemius muscles [84], promote glucose uptake [85], and enhance mitochondrial biosynthesis to improve exercise capacity [86] in animal models. Epidemiological studies have reported that DPP-4i treatment of elderly diabetic patients shows improved sarcopenia parameters compared with other diabetes medications [78], but may not be sufficient to improve non-diabetic sarcopenia.

#### 3.4.5. Glucagon-like Peptide-1 Receptor Agonist (GLP-1 Ras)

GLP-1 RAs bind specifically to GLP-1 receptors and stimulate insulin secretion from pancreatic beta cells. For skeletal muscle, they increase GLUT4 gene expression, glucose uptake, and glycogen synthesis [87]. GLP-1 RAs have been reported to promote skeletal muscle oxygen consumption and increase insulin sensitivity [88]. In a study of diabetic patients, six months of treatment with teneligliptin (20 mg/day), a DPP-4i, decreased the SMI more efficiently than that with dulaglutide (0.75 mg/week), a GLP-1 RA [89]. Although 24 weeks of dulaglutide (3 g/day) treatment in obese elderly patients with diabetes did not show an exacerbation of sarcopenia parameters [90], it is unlikely that GLP-1 RAs improve sarcopenia.

#### 3.4.6. Sodium-Glucose Transporter Protein 2 Inhibitors (SGLT2i)

SGLT2 is expressed in the renal proximal tubules and is responsible for glucose reabsorption. Thus, SGLT2i has a hypoglycemic effect by promoting the excretion of glucose into the urine. For skeletal muscle, long-term administration of SGLT2i has been reported to improve insulin sensitivity, inhibit muscle catabolism, and improve the quality of muscle function [91]. According to a study showing improvement in maximum grip strength, improvement in mitochondrial function is considered to be the molecular mechanism that explains this effect [92]. However, it has been suggested that muscle proteins may be degraded and sarcopenia may occur because of the supply of amino acids to the liver [93]. Results of clinical trials using SGLT2i have shown that it reduces the SMI [94,95], with few reports showing improvement in sarcopenia parameters [96]. Therefore, SGLT2i is unlikely to improve sarcopenia.

#### 3.4.7. Insulin

Insulin acts on skeletal muscle to promote protein synthesis. It also takes up glucose and promotes glycogen synthesis. Anti-insulinemia for younger patients stimulates muscle protein synthesis, but the response is found to be blunted in the elderly [97]. Observational studies have reported that insulin administration prevents a decrease in SMI in diabetic patients [98]. In addition, a 3-year follow-up study of 118 diabetic patients has shown that insulin produces a positive effect on skeletal muscle mass compared with other oral hypoglycemic agents [99]. However, all of these reports were in diabetic patients and would not improve sarcopenia. Figure 2 represents an overview of glucose-lowering drugs for sarcopenia. Table 2 summarizes the pharmacological approach for attenuating muscle atrophy.

## 4. Conclusions

Recent advances in our understanding of muscle biology have revealed molecular mechanisms of sarcopenia and candidate nutritional therapies. Although the combination of resistance training and supplements containing amino acids is usually recommended to prevent age-related muscle weakness and loss [10,11], protein (amino acid)-only supplements had no effect on the symptoms of sarcopenia. Many candidate substances, such as catechins, soy isoflavones, and ursolic acid, have the potential to combat sarcopenia, but even in rodent models of sarcopenia, no systematic and basic research has been conducted on this treatment. Myostatin is a potent inhibitor gene of muscle hypertrophy, and its inhibitors have been tried in many neuromuscular diseases. The effect of myostatin inhibitors on sarcopenia has been positive so far, but do not over-expect it. A more recent scoping review highlights the importance of future research focused on the potential use of Zn in the prevention and management of malnutrition, sarcopenia, and frailty in the elderly [101]. Little is known about whether this micronutrient’s intake improves sarcopenia, which will be an interesting area of research in the future.

## Figures and Tables

**Figure 1 cells-12-02422-f001:**
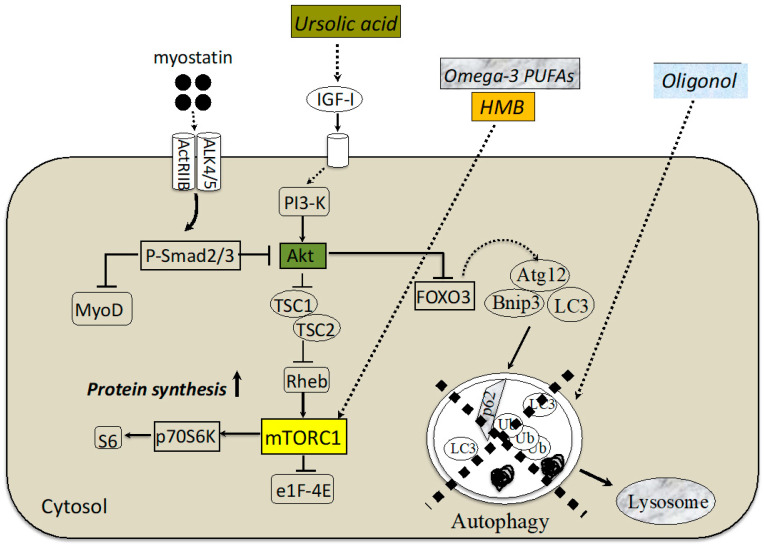
Nutritional interventions affect different mediators in sarcopenic muscle. Recent findings suggest that the myostatin-Smad pathway inhibits protein synthesis probably due to blocking the functional role of Akt. Treatment with an ursolic acid upregulates the amount of IGF-I and then stimulates protein synthesis by activating the Akt/mTORC1/p70S6K pathway. Administration of omega-3 PUFA or HMB may work to prevent sarcopenia by activating mTORC1. Sarcopenic muscle exhibits a marked defect of autophagy-dependent signaling, which is effectively ameliorated by oligonol supplementation. ALK, activin receptor-like kinase; ActRIIB, activin receptor IIB; IGF-I, insulin-like growth factor I; TSC, tuberous sclerosis complex; TORC1, component of TOR signaling complex 1; Rheb, Ras homolog enriched in brain; mTORC1, mammalian target of rapamycin complex 1; eIF4E, eukaryotic initiation factor 4E; FOXO, forkhead box O; LC3, microtubule-associated protein light chain 3; BNIP, BCL2/adenovirus E1B 19 kd-interacting protein; Atg, autophagy-related genes.

**Figure 2 cells-12-02422-f002:**
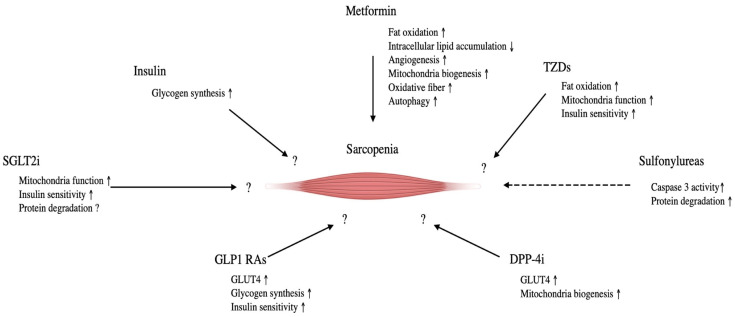
The effect of glucose lowering drugs on sarcopenia and its molecular mechanisms. Tzds, thiazolidinediones; DPP4i, Dipeptidyl pepridaze-4 inhibitors; GLP-1 Ras, Glucogon-like peptide-1 receptor agonist; SLGT2i, Sodium-glucose transporter protein 2 inhibitors. Straight lines indicate a positive effect on sarcopenia; dotted lines indicate negative effects.

**Table 1 cells-12-02422-t001:** Summary of nutritional approaches for attenuating muscle atrophy.

Title 1	Duration, Intervention Pattern	Species	Outcome	Authors
HMB	3 g, 10 days	Human, unloading patients	Muscle strength ↑	Deutz et al. [19]
	2 to 4 weeks	Human	Anthropometric parameters →	Hsieh et al. [20]
	RCT, HMB	Human, elderly	LBM ↑, fat mass → Muscle strength →	Flakoll et al. [21]
HMB	RCT, HMB + Arg + Lys (2/5/1.5 g per day)	Human, elderly	LBM ↑, fat mass → Whole body protein synthesis↑ Leg and grip strength ↑ Limb circumference ↑ Muscle strength →	Flakoll et al. [21]
HMB + vibration	RCT, 12–16 Hz, 3–5 mm, 8 weeks	Human, sarcopenia	Muscle strength ↑, Physical performance (walking speed, 5-stance test, timed stand test) ↑	Zhu et al. [23]
	LMHFV	Ageing mice (SAMP8)	Myostatin expression ↓ Intramuscular fat mass ↓	Wang et al. [24]
Catechin	EGCG, 5 mg/kg, 4 times/week, 8 weeks	Muscular dystrophic mice	fibrosis ↓, necrotic fibers ↓	Nakae et al. [25]
	EGCG, 100 mg/kg, 30 days	Aged male rats	Oxidative stress marker ↓	Senthil Kumaran et al. [26]
	EC or EGCG (0.25% in drinking water), 37 weeks	20-month old male mice	Survival rate: EC ↑, EGCG → Muscle degeneration ↓: EC >> EGCG Physical activity ↑: EC >> EGCG	Si et al. [27]
	Oligonol, 200 mg/kg, 8 weeks	SAMP8 mice, 32-weeks old	PGC-1α and Mfn2 ↓ LC3-II, p62, and ATG13 ↑ Autophagosome number ↑	Chang et al. [30]
Catechin + resistance training	RCT, tea catechin (350 mL/day), 3 month	Human, sarcopenia, women	Leg muscle mass ↑ Normal walking speed ↑	Mafi et al. [28]
	120 days	Male mice	Fat accumulation ↓	Kurrat et al. [32]
	20% of the diet, 4 days	Mouse, denervated	IRS-1 and p-Akt protein ↑	Abe et al. [33]
Isoflavones	0.6% of the diet, 2 weeks daidzein/genistein/ glycitein (7:1:2)	Mouse, denervated	Apoptosis-dependent signaling ↓	Tabata et al. [34]
	Muscular injection (50 μL; 2.5 pmol of quercetin)	Hindlimb-unloaded mice	Skeletal muscle mass ↑ Atrogin-1 and MuRF1 ↓	Mukai et al. [35]
Ursolic acid	Orally treatment (100 mg/kg), 3 weeks	Mice models of CKD	Muscle mass ↑ Inflammatory cytokines (IL-6 and TNF-α) ↓ Ubiquitin E3 ligases (MuRF1, atrogin-1, MUSA1) ↓	Yu et al. [37]
	50 mg/kg, loquat leaf extract intake	Human (healthy adults)	Muscle mass and muscle strength →	Cho et al. [39]
Ursolic acid + endurance training	500 mg/kg, 8 weeks	21-month old male rats with diabetes by high fat diet	Body weight ↓ Insulin resistance ↓ Insulin and glucose concentration ↓ p53, ATF4, and p21 protein level ↓	Zolfaghari et al. [38]
Ursolic acid + resistance training	1 capsule (450 mg), 3 times/day, 8 weeks	Healthy male human (age: 29.4 ± 5.1 years)	Muscle strength ↑ Lean body mass →	Bang et al. [40]
	1.86 g EPA and 1.5 g DHA/day, 6 months	Human (60-85 year men and women, elderly)	Muscle protein synthesis ↑	Smith et al. [43]
Omega-3 PUFAs	RCT, EPA and DHA, 8 weeks	Middle-aged and older adults	TNF-α, IL-6, and IL-1β ↓	Tan et al. [45]
	RCT, 8 weeks	Healthy older men and women (>65 years)	Muscle protein synthesis →	Smith et al. [46]

↑: increase, ↓: decrease, → no change.

**Table 2 cells-12-02422-t002:** Summary of pharmacological approach for attenuating muscle atrophy.

Title 1	Duration, Intervention Pattern	Species	Outcome	Authors
Vitamin D	Systematic review	Nursing home residents and older adults living in areas with low vitamin D	Risk of falls ↓ Muscle performance ↑ Muscle strength ↑	Annweiler et al. [47]
	Subcutaneous injection	CKD model mice	Body weight ↑ Muscle mass ↑ Inflammation marker ↓	Zhang et al. [52]
Myostatin inhibition	RCT, ACE-031, Cohort 1: 0.5 mg/kg every 4 weeks, Cohort 2: 1 mg/kg every 2 weeks	DMD patients	Muscle mass ↑ Performance (6 min walk test) ↑ Non-muscular side effects (nosebleeds, capillary dilation) ↑	Campbell et al. [55]
	RCT, LY2495655, 8 and 16 weeks	Patients undergoing elective total hip arthroplasty	LBM ↑	Woodhouse et al. [56]
	Phase 2 trial, LY2495655, 24 weeks	Frail elderly (75 years and older) subjects worldwide (e.g., US, France, Australia)	LBM ↑ Functional characteristics (stair climbing time and chair standing with arms) ↑	Becker et al. [57]
GDF11	IP injection, 0.1 mg/kg, 30 days	Aged (22–24 months) mice	Satellite cells ↑ Skeletal muscle regeneration ↑ Muscle physiological parameters (Run time and grip strength) ↑ PGC-1α ↑	Sinha et al. [60]
Anabolic steroids	Testosterone, high dose	Community-dwelling older men, 6 months	LBM ↑ Leg and arm muscle strength ↑	Sinha-Hikim et al. [67]
	Double-blind study, SARMs (enobosarm)	Healthy postmenopausal elderly men	LBM ↑ Physical function (stair climbing speed) ↑	Neil et al. [70]
	SARMs (GSK2881078), Once daily, 50 days	Healthy elderly men and women	LBM ↑	Neil et al. [70]
	RCT, oxymetholone, 24-weeks	Hemodyalysis patients	LBM ↑ Physical function and grip strength ↑ CSA of Type I fibers ↑	Supasyndh et al. [71]
	Oxymetholone, 50 mg/kg	10-month-old mice	Myostatin, sirtuin1, Fiber size (soleus and gastrocnemius muscles) ↑	Kim et al. [72]
	Metformin, 3 × 500 mg, 16 weeks	Elderly patients with diabetes	Walking speed ↑	Laksmi et al. [77]
	Thiazolidinediones (pioglitazone)	Diabetic patients	Thigh muscle mass ↓	Shea et al. [80]
	Sulfonylurea, 24 month	Elderly patients with diabetes	Skeletal muscle mass → Muscle strength → Gait speed →	Rizzo et al. [82]
Glucose-lowering drugs	DPP-4i, 24 month	Elderly patients with diabetes	Skeletal muscle mass ↑ Muscle strength ↑ Gait speed ↑	Rizzo et al. [82]
	DPP-4i (tenelingliptin), 20 mg/day, 6 months	Diabetic patients undergoing hemodyalysis	Skeletal muscle mass → Fat mass →	Yajima et al. [89]
	GLP-1 RAs (dulaglutide), 0.75 mg/week, 6 months	Diabetic patients undergoing hemodyalysis	Skeletal muscle mass ↓ Fat mass ↓	Yajima et al. [89]
	GLP-1 RAs (dulaglutide), 3 g/day, 6 months	Obese elderly patients with diabetes	Sarcopenic parameters →	Perna et al. [90]
	SGLT2i (depaglifrozin), 10 mg/day, 2 weeks	Diabetic patients	Insulin sensitivity ↑ Muscle catabolism ↓	Merovci et al. [91]
	SGLT2i (luseoglifrozin), 2.5–5 mg/day, 1 year	Diabetic patients	Body weight ↓ Body mass index ↓ Waist circumference ↓	Sasaki et al. [100]
	Insulin, 3 years	Diabetic patients	Skeletal muscle mass ↑	Ferrari et al. [99]

↑: increase, ↓: decrease, → no change.

## Data Availability

We confirm that our MS excluded unpublished data and figures.

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
