# Peer review of "Current Nutritional and Pharmacological Approaches for Attenuating Sarcopenia"

_cells, 2023, doi:10.3390/cells12192422_

Round 1
Reviewer 1 Report
The theme of the manuscript is a topical one and represents a problem that affects the quality of life, especially for the elderly. However, some improvements are necessary that I propose:
- it is important in a review article to make a brief presentation of the informative database used to create the manuscript;
- for a better presentation of the information, I suggest the authors to present a series of systematized data in a table;
- maybe it is good to include information about the role of some micronutrients in improving muscle mass (for example Cu and Zn);
- also, for certain active substances (ursolic acid for example) for better information, more sources should be specified (not just apples).
Author Response
The theme of the manuscript is a topical one and represents a problem that affects the quality of life, especially for the elderly. However, some improvements are necessary that I propose:
>>> Thank you for kind opinion for my review article.
- it is important in a review article to make a brief presentation of the informative database used to create the manuscript;
>>> We added the explanation of the informative database (page 1, line 64).
- for a better presentation of the information, I suggest the authors to present a series of systematized data in a table;
>>> Thank you for important opinion. We added two tables (Table 1 & 2) describing a series of systematized data.
- maybe it is good to include information about the role of some micronutrients in improving muscle mass (for example Cu and Zn);
>>> Thank you for good suggestion. We added the explanation for a possible role of micronutrients improving sarcopenia. (page 13, lines 444-447).
- also, for certain active substances (ursolic acid for example) for better information, more sources should be specified (not just apples).
>>>We added more information for certain active substances (page 4, lines 163-164).
Reviewer 2 Report
This is a useful contribution to the literature, in reviewing nutritional and pharmacological approaches to attenuating sarcopenia, from a basic science point of view. Yet the article aspires to comment on clinical studies in humans in the abstract. It cites whole body vibration stimulation when commenting on evidence about exercise.
If the authors really wish to comment on the need for human and clinical studies, then the whole body of clinical research has been omitted. Please refer to some of the more recent summaries in this field. If the authors do not wish to amend the MS to be a more balanced review, then it should state explicitly that this is a review about basic mechanisms and delete all mention about the need for human studies etc, but mention that there has been a very active clinical research field in the areas of sarcopenia for the last 20 years, more recently including studies of muscle physiology.
Three articles for reference:
Dent E, Morley JE, Cruz-Jentoft AJ, Arai H, Kritchevsky SB, Guralnik J, Bauer JM, Pahor M, Clark BC, Cesari M, Ruiz J, Sieber CC, Aubertin-Leheudre M, Waters DL, Visvanathan R, Landi F, Villareal DT, Fielding R, Won CW, Theou O, Martin FC, Dong B, Woo J, Flicker L, Ferrucci L, Merchant RA, Cao L, Cederholm T, Ribeiro SML, Rodríguez-Mañas L, Anker SD, Lundy J, Gutiérrez Robledo LM, Bautmans I, Aprahamian I, Schols JMGA, Izquierdo M, Vellas B. International Clinical Practice Guidelines for Sarcopenia (ICFSR): Screening, Diagnosis and Management. The Journal of Nutrition, Health & Aging 2018; 22(10): 1148-1161.
Zhu LY, Chan R, Kwok T, Cheng KCC, Ha A, Woo J. Effects of exercise and nutrition supplementation in community-dwelling older Chinese people with sarcopenia: a randomized controlled trial. Age and Ageing 2019; 48: 220-228.
Chen LK, Arai H, Assantachai P, Akishita M, Chew STH, Dumlao LC, Duque G, Woo J. Roles of nutrition in muscle health of community‐dwelling older adults: evidence‐based expert consensus from Asian Working Group for Sarcopenia. Journal of Cachexia, Sarcopenia and Muscle 2022; 13(3): 1653-1672.

Author Response
This is a useful contribution to the literature, in reviewing nutritional and pharmacological approaches to attenuating sarcopenia, from a basic science point of view. Yet the article aspires to comment on clinical studies in humans in the abstract. It cites whole body vibration stimulation when commenting on evidence about exercise.
If the authors really wish to comment on the need for human and clinical studies, then the whole body of clinical research has been omitted. Please refer to some of the more recent summaries in this field. If the authors do not wish to amend the MS to be a more balanced review, then it should state explicitly that this is a review about basic mechanisms and delete all mention about the need for human studies etc, but mention that there has been a very active clinical research field in the areas of sarcopenia for the last 20 years, more recently including studies of muscle physiology.
>>> Thank you for very important suggestion. We referred following 2 articles (Dent E et al., 2018, Chen et al., 2022). We added explanations concerning supplementation of protein (amino acids), vitamin D, omega-3, and pharmaceuticals for sarcopenia in human according to recent ICFSR and AWGS guidelines. [page 2, lines 68-72; page 5, lines 188-192; page 7, lines 251-252; page 8, lines 271-272 and 314-315]
Three articles for reference:
Dent E, Morley JE, Cruz-Jentoft AJ, Arai H, Kritchevsky SB, Guralnik J, Bauer JM, Pahor M, Clark BC, Cesari M, Ruiz J, Sieber CC, Aubertin-Leheudre M, Waters DL, Visvanathan R, Landi F, Villareal DT, Fielding R, Won CW, Theou O, Martin FC, Dong B, Woo J, Flicker L, Ferrucci L, Merchant RA, Cao L, Cederholm T, Ribeiro SML, Rodríguez-Mañas L, Anker SD, Lundy J, Gutiérrez Robledo LM, Bautmans I, Aprahamian I, Schols JMGA, Izquierdo M, Vellas B. International Clinical Practice Guidelines for Sarcopenia (ICFSR): Screening, Diagnosis and Management. The Journal of Nutrition, Health & Aging 2018; 22(10): 1148-1161.
Zhu LY, Chan R, Kwok T, Cheng KCC, Ha A, Woo J. Effects of exercise and nutrition supplementation in community-dwelling older Chinese people with sarcopenia: a randomized controlled trial. Age and Ageing 2019; 48: 220-228.
Chen LK, Arai H, Assantachai P, Akishita M, Chew STH, Dumlao LC, Duque G, Woo J. Roles of nutrition in muscle health of community‐dwelling older adults: evidence‐based expert consensus from Asian Working Group for Sarcopenia. Journal of Cachexia, Sarcopenia and Muscle 2022; 13(3): 1653-1672.
Reviewer 3 Report
The review of K. Sakuma et al. draws attention to current nutritional and pharmacological approaches to counteract or mitigate sarcopenia. The subject of paper is very interesting and topical, moreover it has a considerable social-health importance, particularly as sarcopenia is often promoted by nutritional deficiencies. In fact, in physiological conditions, the nutritional approach probably represents the best way to prevent or limit sarcopenia. Regarding the nutritional approach, the authors focus solely on HMB, polyphenols, UA and Omega-3. The work is well written and easy to read.
The conclusions are quite in line with the text, however the reference to supplementation with amino acids and proteins (lines 444-447) was not taken into account in the text. The authors, rightly, state that the combination of resistance exercise along with amino acid supplementation is the normally recommended strategy to prevent muscle loss. There is a substantial literature on the role of mixtures containing all essential amino acids (EAA) on muscle metabolism and on the prevention of sarcopenia, both in experimental models and in humans. Then, I would invite the authors to introduce an initial paragraph into the text which concerns precisely this aspect. In fact, the adequate availability of all EAAs is a necessary condition at the basis of an efficient cellular metabolism, even more so if in the presence of a hypercatabolic state. Similarly, regarding the statement on proteins at lines 446-447.
Author Response
The review of K. Sakuma et al. draws attention to current nutritional and pharmacological approaches to counteract or mitigate sarcopenia. The subject of paper is very interesting and topical, moreover it has a considerable social-health importance, particularly as sarcopenia is often promoted by nutritional deficiencies. In fact, in physiological conditions, the nutritional approach probably represents the best way to prevent or limit sarcopenia. Regarding the nutritional approach, the authors focus solely on HMB, polyphenols, UA and Omega-3. The work is well written and easy to read.
>>> Thank you for kind opinion for my review article.
The conclusions are quite in line with the text, however the reference to supplementation with amino acids and proteins (lines 444-447) was not taken into account in the text. The authors, rightly, state that the combination of resistance exercise along with amino acid supplementation is the normally recommended strategy to prevent muscle loss. There is a substantial literature on the role of mixtures containing all essential amino acids (EAA) on muscle metabolism and on the prevention of sarcopenia, both in experimental models and in humans. Then, I would invite the authors to introduce an initial paragraph into the text which concerns precisely this aspect. In fact, the adequate availability of all EAAs is a necessary condition at the basis of an efficient cellular metabolism, even more so if in the presence of a hypercatabolic state. Similarly, regarding the statement on proteins at lines 446-447.
>>> Thank you for good suggestion. We exchanged several articles concerning the combination of resistance exercise along with amino acid supplementation for sarcopenia (page 2, Ref. 10 & 11). We added several explanations and references for the importance of essential amino acids on muscle metabolism and on the prevention of sarcopenia (page 2, lines 53-59).
Round 2
Reviewer 1 Report
Accept in present form
Reviewer 3 Report
the text has been improved by offering a broader overview of the topic addressed by the authors